# Volume-Controlled Versus Dual-Controlled Ventilation during Robot-Assisted Laparoscopic Prostatectomy with Steep Trendelenburg Position: A Randomized-Controlled Trial

**DOI:** 10.3390/jcm8122032

**Published:** 2019-11-21

**Authors:** Jin Ha Park, In Kyeong Park, Seung Ho Choi, Darhae Eum, Min-Soo Kim

**Affiliations:** 1Department of Anesthesiology and Pain Medicine, and Anesthesia and Pain Research Institute, Yonsei University College of Medicine, Seoul 03722, Korea; realsummer@yuhs.ac (J.H.P.); csho99@yuhs.ac (S.H.C.); deum@yuhs.ac (D.E.); 2Department of Anesthesiology, College of medicine, Kangwon national university, Chuncheon 24341, Korea; orbitless@naver.com

**Keywords:** arterial oxygenation, dual-controlled ventilation, respiratory mechanics, robot-assisted laparoscopic radical prostatectomy, volume-controlled ventilation

## Abstract

Dual-controlled ventilation (DCV) combines the advantages of volume-controlled ventilation (VCV) and pressure-controlled ventilation (PCV). Carbon dioxide (CO_2_) pneumoperitoneum and steep Trendelenburg positioning for robot-assisted laparoscopic radical prostatectomy (RALRP) has negative effects on the respiratory system. We hypothesized that the use of autoflow as one type of DCV can reduce these effects during RALRP. Eighty patients undergoing RALRP were randomly assigned to receive VCV or DCV. Arterial oxygen tension (PaO_2_) as the primary outcome, respiratory and hemodynamic data, and postoperative fever rates were compared at four time points: 10 min after anesthesia induction (T1), 30 and 60 min after the initiation of CO_2_ pneumoperitoneum and Trendelenburg positioning (T2 and T3), and 10 min after supine positioning (T4). There were no significant differences in PaO_2_ between the two groups. Mean peak airway pressure (Ppeak) was significantly lower in group DCV than in group VCV at T2 (mean difference, 5.0 cm H_2_O; adjusted *p* < 0.001) and T3 (mean difference, 3.9 cm H_2_O; adjusted *p* < 0.001). Postoperative fever occurring within the first 2 days after surgery was more common in group VCV (12%) than in group DCV (3%) (*p* = 0.022). Compared with VCV, DCV did not improve oxygenation during RALRP. However, DCV significantly decreased Ppeak without hemodynamic instability.

## 1. Introduction

Recent advances in technology have led to the availability of new ventilation modes combining the advantages of both volume-controlled ventilation (VCV) and pressure-controlled ventilation (PCV) in several new anesthesia machines. These new modes, known as dual-controlled ventilation (DCV) modes, include PCV with volume guaranteed (PCV-VG; General Electric), pressure-regulated volume control (PRVC; Maquet), and volume mode with autoflow (AF; Dräger), depending on the manufacturer [1]. DCV modes aim to deliver a constant tidal volume using a decelerating flow pattern with a constant inspiratory pressure. The decelerating flow provides the higher instantaneous peak inspiratory flow and may lead to a better alveolar recruitment. Hence, the use of DCV may reduce inspiratory pressure, as well as atelectasis [2,3].

Carbon dioxide (CO_2_) pneumoperitoneum and steep Trendelenburg positioning are commonly used in combination to provide adequate surgical viewing and space during robot-assisted or laparoscopic abdominal surgeries. However, they are likely to produce significant and potentially negative physiological changes in various organ systems, including the respiratory system [4,5]. Increased intra-abdominal pressure decreases lung compliance and functional residual capacity [6,7,8]. Resulting atelectasis can reduce arterial oxygenation, and higher peak and plateau inspiratory pressure may increase the risk of barotrauma [5,9,10].

Robot-assisted laparoscopic radical prostatectomy (RALRP) is a commonly performed robotic surgery using CO_2_ pneumoperitoneum and steep Trendelenburg positioning, which is usually performed in elderly patients with reduced cardiopulmonary reserve. Considering the aforementioned adverse respiratory changes during RALRP and the decline in pulmonary function in the elderly, DCV modes might improve arterial oxygenation and prevent lung injury due to high inspiratory pressure [2,11].

We hypothesized that application of DCV which ensures tidal volume with a decelerating flow pattern could be beneficial to improve oxygenation and prevent barotrauma by reducing negative effects of CO_2_ pneumoperitoneum and steep Trendelenburg position on the respiratory system during RALRP. Therefore, we designed this study to compare the effects of DCV and VCV mode on oxygenation and airway pressure in patients undergoing RALRP.

## 2. Experimental Section

The Institutional Review Board of Severance Hospital, Yonsei University Health System, Seoul, Republic of Korea, approved this prospective, randomized, parallel-group, double-blinded, unfunded, single-center trial (4-2016-0311), and the protocol was registered at http://clinicaltrials.gov (ref: NCT02803424). This study was conducted between June 2016 and December 2016 at Severance Hospital.

### 2.1. Patients

After written informed consent was obtained from eligible patients, 80 adult male patients scheduled for elective RALRP using the da Vinci^TM^ Robot system (Intuitive Surgical, Inc., Mountain View, CA, USA) were enrolled in the study. We excluded patients with heart failure, asthma, history of airway hyperresponsiveness in previous anesthesia, chronic obstructive pulmonary disease (COPD), findings of preoperative chest X-ray or pulmonary function test that were suspicious of COPD, and obesity (body mass index (BMI) > 30 kg∙m^−2^).

The enrolled patients were allocated randomly according to a predetermined randomization list to receive either VCV mode (group VCV) or DCV (group DCV) mode during CO_2_ pneumoperitoneum and steep Trendelenburg positioning. The randomization list was generated from an Internet website (http://www.random.org/) with an allocation ratio of 1:1 and no blocking, and a paper containing the group identification information was inserted in an opaque envelope. Patient enrolment and group allocation were conducted by a researcher who was not involved in the anesthesia or data analysis. Independent anesthesiologists familiar with robot-assisted surgeries provided anesthesia in a standardized manner. The attending anesthesiologists and outcome investigators were aware of the group assignment, but the patients, surgeons, and data analysts were blinded to the allocated group.

### 2.2. Anesthetic Management

Routine monitoring devices [12], including electrocardiography, a non-invasive arterial blood pressure measurement and pulse oximetry, were applied on arrival at the operating room. Intravenous propofol 1.5 mg∙kg^−1^ and desflurane at an end-tidal concentration of 5%–6% with 100% oxygen were administered for anesthesia induction. Intravenous rocuronium 0.6 mg∙kg^−1^ was administered with a continuous infusion of remifentanil at a rate of 0.1–0.2 μg∙kg^−1^∙min^−1^ to facilitate endotracheal intubation. Remifentanil infusion, desflurane, and a continuous infusion of rocuronium 0.6 μg∙kg^−1^∙h^−1^ was performed during the intraoperative period [13]. After induction, the radial artery was cannulated for continuous blood pressure (BP) monitoring, as well as blood sampling for arterial blood gas analysis (ABGA). The remifentanil infusion rate was adjusted to maintain the mean arterial pressure (MAP) and heart rate (HR) within approximately 20% of baseline. The inhaled concentration of desflurane was adjusted to maintain the bispectral index score (BIS) (A-2000 BIS Monitor^TM^; Aspect Medical System Inc., Newton, MA, USA) between 40 and 60. The volume of fluid administered during surgery and use of vasoactive drugs (including ephedrine, phenylephrine, and norepinephrine) were left to the discretion of the attending anesthesiologist. Throughout surgery, a forced-air warming system (Bair-Hugger^TM^; Augustine-Medical, Eden Prairie, MN, USA) was used to keep the body temperature at 36.0–37.0 °C. At the end of the operation, all anesthetic agents were discontinued, and sugammadex 4 mg∙kg^−1^ was administered to reverse residual neuromuscular blockade [13].

### 2.3. Ventilation Management

The study flow chart is shown in Figure 1. Immediately after induction, all patients were ventilated with VCV mode (Primus ventilator, Dräger Medical, Lübeck, Germany) with an inspiratory to expiratory (I/E) ratio of 1:2; tidal volume of 8 mL∙kg^−1^ ideal body weight, the duration of inspiratory pause (i.e., the plateau time set to 10% of total inspiration time); no positive end-expiratory pressure (PEEP); and 2 L∙min^−1^ oxygen/air mixtures set to deliver 50% fraction of inspiratory oxygen (FiO_2_). Ideal body weight (in kg) was calculated using the following formula for men: 50 + 0.91(height (cm) − 152.4) [14]. The respiratory rate (RR) was controlled to maintain a mean (standard deviation (SD)) end-tidal carbon dioxide (EtCO_2_) in the range of 40 (5) mmHg during CO_2_ pneumoperitoneum with the Trendelenburg position, so the plateau time in this study varied according to the RR.

Initially, CO_2_ pneumoperitoneum was achieved with an intra-abdominal pressure of 15 mmHg while the patient was in the supine position. A 30° Trendelenburg position was then established, which was verified by the surgical bed controller that displayed the set angle in numerical form. Immediately after Trendelenburg positioning with CO_2_ pneumoperitoneum, the ventilator mode in the DCV group was changed from VCV to DCV without changes in settings of other ventilatory parameters. After completion of the robot-assisted procedure, the supine position was resumed, CO_2_ was desufflated, and the ventilator mode in group DCV was changed back to VCV. In the group VCV, VCV mode was maintained throughout the surgery.

### 2.4. Data Collection

ABGA, respiratory data, and hemodynamic data were collected at four times: 10 min after anesthesia induction under the supine position and no pneumoperitoneum (T1); 30 min after initiation of CO_2_ pneumoperitoneum and Trendelenburg positioning (T2); 60 min after initiation of CO_2_ pneumoperitoneum and Trendelenburg positioning (T3); and 10 min after CO_2_ desufflation and supine positioning (T4). If the robot-assisted procedure was completed before T3, data corresponding to T3 were collected at the time of completion, before the CO_2_ pneumoperitoneum and Trendelenburg positioning were discontinued. ABGA data included arterial pH, arterial oxygen tension (PaO_2_), arterial CO_2_ tension (PaCO_2_), and lactate level. Hemodynamic data included BP and HR. Respiratory data included peak airway pressure (Ppeak), plateau airway pressure (Pplat), mean airway pressure (Pmean), EtCO_2_, and RR, which were collected by reading the values displayed on the monitor of the Primus anesthetic workstation (Dräger, Lübeck, Germany) once at each time point. We also collected data regarding the duration of surgery, pneumoperitoneum, anesthesia, post-anesthesia care unit (PACU) stay, and hospital stay; intraoperative blood loss, fluid intake, urine output, amount of administered vasopressors, and respiratory complications, such as pneumothorax and desaturation events during intraoperative and recovery period; and respiratory complications and presence of postoperative fever (>38.0 °C) within two postoperative days. To compare vasopressor usage in each group, the amount of administered phenylephrine was converted to ephedrine equivalents using a relative potency ratio for phenylephrine/ephedrine of 80:1 [15]. According to our study protocol, patients were withdrawn from the study if oxygen desaturation (pulse oximetry (SpO_2_) < 95%) or increase of Ppeak > 40 cmH_2_O occurred [9].

### 2.5. Statistical Analysis

Statistical analysis was conducted with R version 3.4.1 (R Foundation for Statistical Computing, Vienna, Austria). Assuming a power of 80% and an alpha level of 0.05, our sample size calculations indicated that 32 patients per group were required to detect a difference of 25 mmHg (approximately 16% of the mean value) in PaO_2_ measured from ABGA data 30 min after initiation of CO_2_ pneumoperitoneum and Trendelenburg positioning (T2), which was our primary outcome [9]. Considering patient dropout, we aimed to enroll 40 patients per group.

Patient characteristics and perioperative data between the two groups were compared with the chi-squared test or Fisher’s exact test for categorical data and the independent *t*-test or Mann–Whitney U-test for continuous variables. Repeatedly measured outcomes (including hemodynamic, respiratory, and ABGA data) at each time point were analyzed using a linear mixed model with patient indicator as a random effect, and group, time, and group-by-time as fixed effects. This was followed by post hoc analysis for multiple comparisons using the Bonferroni correction. All statistical analyses were conducted with two-tailed tests, and all *p* or Bonferroni-corrected *p* values < 0.05 were considered statistically significant.

## 3. Results

We initially assessed 81 patients for eligibility. One refused to participate in the study, leaving 80 men who were enrolled and randomized. Two patients from each group were withdrawn from the analysis because of high Ppeak and protocol violations; intra-abdominal pressure had to be reduced from 15 mmHg to 10 mmHg due to excessively elevated Ppeak, and errors occurred in the setting of FiO_2_ and tidal volume on the ventilator machine. Therefore, 76 patients were available for analysis (Figure 2). Patient characteristics and perioperative data were similar between the two groups (Table 1).

ABGA and EtCO_2_ data at each time point are shown in Table 2. For the primary outcome, there was no significant difference in PaO_2_ at T2 between the two groups (mean difference, −12.7 mmHg; 95% confidence interval (CI) −27.9 to 2.5; adjusted *p* = 0.400) (Figure 3A). The mean PaO_2_ at this time was higher in DCV group than in VCV group, but the difference did not reach statistical significance. All other ABGA and EtCO_2_ data were similar between the two groups.

Respiratory data are shown in Table 3. Significant interactions of group and time were observed in linear mixed model analysis for Ppeak, Pplat, and Pmean. Post hoc analysis revealed significantly lower mean Ppeak in DCV group than in VCV group at T2 (mean difference, 5.0 cm H_2_O; 95% CI, 3.5 to 6.5; adjusted *p* < 0.001) and T3 (mean difference, 3.9 cm H_2_O; 95% CI, 2.2 to 5.5; adjusted *p* < 0.001) (Figure 3B). Pplat was significantly lower in DCV group than in VCV group at T2 in post hoc analysis (mean difference, 2.9 cm H_2_O; 95% CI, 1.2 to 4.7; adjusted *p* = 0.004). There were no significant differences in Pmean between the two groups in post hoc analysis.

Hemodynamic data were comparable between the two groups (Table 4). The incidence of postoperative fever was significantly higher in VCV group (12%) than in DCV group (3%) (*P* = 0.022). The fever occurred on the day of surgery or the first or second postoperative days, was benign, and resolved with antipyretics in all patients. No patient exhibited respiratory complications during the intraoperative or postoperative periods.

## 4. Discussion

In this study, DCV did not provide significantly better oxygenation than VCV. However, DCV was associated with significantly decreased Ppeak without hemodynamic instability during CO_2_ pneumoperitoneum and Trendelenburg positioning in patients undergoing RALRP. In addition, postoperative fever was significantly less frequent in patients receiving DCV than in those receiving VCV.

During laparoscopic surgery, the elevated intra-abdominal pressure induces abdominal wall distension, an increase of the abdominal wall elastance, and a cranial shift of the diaphragm [16]. The Trendelenburg positioning allows the weight of abdominal contents to be transmitted to the lung parenchyma [5,6]. In terms of respiratory mechanics, both CO_2_ pneumoperitoneum and Trendelenburg positioning increases chest wall and lung elastance, concomitantly decreasing transpulmonary pressure, which is the pressure that actually expands the lung during positive pressure ventilation [16].

Various strategies, such as the use of PEEP, PCV, a prolonged I/E ratio, and recruitment maneuvers, have been investigated with the purpose of reducing airway pressure and improving oxygenation during CO_2_ pneumoperitoneum and Trendelenburg positioning during RALRP [9,17,18,19,20]. Previous studies demonstrated that use of PEEP was associated with improved oxygenation during RALRP [17,19]; however, the possibility of excessive increases in airway pressure should be considered when applying the PEEP, especially at 10 cmH_2_O or higher [19]. PCV is characterized by an increased mean distribution time because of a decelerating flow pattern with a high initial flow rate [1,20]. These features might allow lower airway pressures and improved gas exchange in clinical situations with an increased risk of lung injury [20,21]. In a randomized trial comparing PCV and VCV during RALRP, PCV provided significantly lower Ppeak than VCV, but no superiority over VCV in terms of arterial oxygenation [20]. In this previous trial, the lack of oxygenation improvement with PCV was explained by similar Pmean between the two modes [20]. In a patient receiving positive pressure ventilation, Pmean is closely related to mean alveolar pressure—the average pressure at the level of the alveoli that functions to open and inflate alveoli against the combined elastic recoil of the chest wall and lung [22,23]. Thus, an increased Pmean may enhance arterial oxygenation by recruiting collapsed alveoli and reducing shunt [9,20,24]. The typical ventilatory strategy for increasing Pmean is the prolonged I/E ratio ventilation [22]. In recent trials comparing the 1:1 equal ratio ventilation (ERV) and the 1:2 conventional ratio ventilation (CRV) during RALRP, PaO_2_ did not differ significantly between the two modes of ventilation, although a significantly increased Pmean was observed in ERV [20]. However, a recent meta-analysis concluded that ERV improved arterial oxygenation at 20–30 min and 60 min after laparoscopy. In addition, the ERV provided improved oxygenation at 60 min after initiation of one-lung ventilation [10].

In the current study, the lack of improvement in arterial oxygenation with DCV may be attributed to comparable Pmean values with DCV and VCV because DCV uses an exponential decay pattern of inspiratory flow, similar to PCV. In a similar surgical setting (i.e., 30° Trendelenburg position with pneumoperitoneum), PCV-VG as a type of DCV mode also did not significantly increase Pmean and PaO_2_ compared with VCV [11]. Thus, the relationship between Pmean and PaO_2_ in DCV mode remains unclear. To date, this mode appears to provide no distinct advantages, such as improved oxygenation, other than a consistently reduced Ppeak. Surgical conditions or patient factors such as age, underlying disease, nutrition status, and functional limitation may be more important than respiratory mechanics in determining oxygenation during DCV or PCV-VG [25,26]. Further randomized studies and meta-analysis are necessary to clarify the effects of DCV mode on arterial oxygenation in various surgical settings.

Although the actual effects of DCV on clinical outcomes are still not established, the absence of requirement for inspiratory pressure adjustment to maintain a predetermined tidal volume is an obvious clinical benefit of using DCV in surgical conditions where lung compliance changes frequently [1]. In RALRP, frequent changes in lung compliance may occur due to surgical manipulations [9]. When a sudden decrease in lung compliance occurs, VCV may not provide sufficient ventilation to the patient because delivering a desired tidal volume results in high airway pressures. PCV under the identical inspiratory pressure delivers various tidal volume depending on changes of lung compliance, and abruptly increased lung compliance may, thus, provoke the delivery of large tidal volume to the patient. Given that DCV delivers a predetermined tidal volume with the lowest possible inspiratory pressure, DCV may reduce the risk of hypoventilation or the delivery of potentially harmful large tidal volumes due to variations in lung compliance during RALRP [1]. The major concern with using a prolonged I/E ratio is the possibility of adverse consequences, such as decreased cardiac output, barotrauma, and air-trapping, due to intrinsic PEEP and a sustained increase of Pmean [10]. Given the difficulty in identifying and quantifying intrinsic PEEP under surgical settings, use of a prolonged I/E ratio may be limited in patients at risk of alveolar rupture, such as those with chronic obstructive lung disease [10,27]. The application of DCV might be beneficial in these patients because it lowers Ppeak by ensuring a sufficient duration for expiration, and thereby preventing incomplete exhalation.

Early postoperative fever is common after many surgical procedures [28,29]. Atelectasis has been known as the primary cause of early postoperative fever [30]. In this current study, DCV was associated with a significantly lower incidence of postoperative fever. However, early postoperative fever may be benign, and unrelated to the surgical procedure [29]. A recent study demonstrated that there was no obvious evidence supporting atelectasis as the main cause of early postoperative fever [28]. Instead, surgical stress and the accompanying increase in circulating inflammatory cytokines may be associated with early postoperative fever [28,31]. The different ventilator modes may have affected surgical stress and the inflammatory response, but this issue was beyond the scope of the current study.

Our study had some limitations, which require consideration. First, the attending anesthesiologists and outcome investigators could not be blinded because of the study design. Second, we did not enrol patients with obesity, respiratory diseases, or heart failure, because these patients might be vulnerable to serious deterioration of respiratory mechanics and oxygenation during CO_2_ pneumoperitoneum and steep Trendelenburg positioning [9,10,20]. Lastly, patients were ventilated without applying PEEP to exclusively compare VCV and DCV. The primary effect of PEEP consists of an increase in functional residual capacity caused by the expansion and stabilization of collapsed alveoli. Therefore, the application of adequate PEEP may result in different contributions to oxygenation and lung compliance during CO_2_ pneumoperitoneum and steep Trendelenburg positioning. Thus, further studies are needed to identify the role of the PEEP and its contribution to oxygenation under each ventilatory mode.

## 5. Conclusions

In conclusion, our study did not support the application of DCV for improving oxygenation. However, this study showed that DCV has a potential to lower Ppeak and Pplat without producing hemodynamic instability during CO_2_ pneumoperitoneum with steep Trendelenburg positioning.

## Figures and Tables

**Figure 1 jcm-08-02032-f001:**
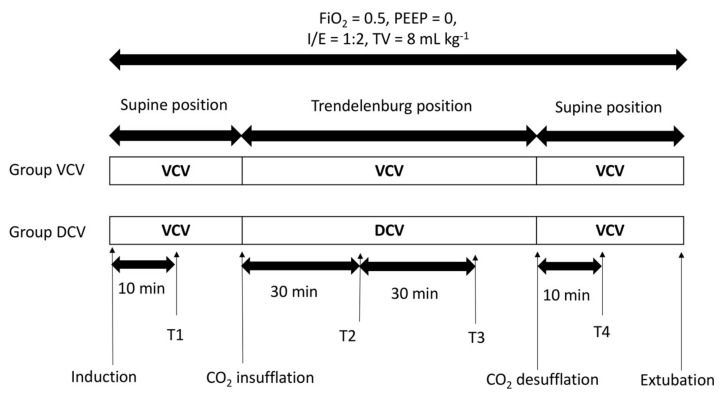
Study flow chart. Abbreviations: FiO_2_ = fraction of inspiratory oxygen; PEEP = positive end-expiratory pressure; I/E, inspiratory to expiratory time; TV = tidal volume; VCV = volume-controlled ventilation; DCV = dual-controlled ventilation. Note: T1 = 10 min after anesthesia induction under supine positioning and no pneumoperitoneum; T2 = 30 min after initiation of CO_2_ pneumoperitoneum and Trendelenburg positioning; T3 = 60 min after initiation of CO_2_ pneumoperitoneum and Trendelenburg positioning; T4 = 10 min after CO_2_ desufflation and supine positioning.

**Figure 2 jcm-08-02032-f002:**
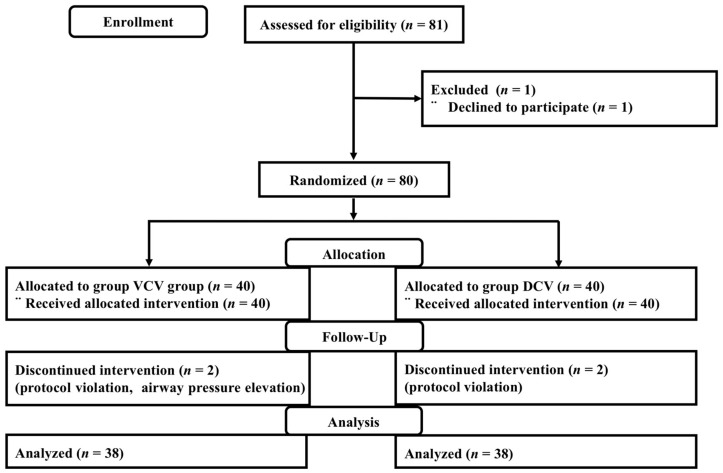
Patient enrolment into the study (using CONSORT recommendations). Abbreviations: VCV = volume-controlled ventilation; DCV = dual-controlled ventilation, CONSORT = consolidated standards of reporting trial.

**Figure 3 jcm-08-02032-f003:**
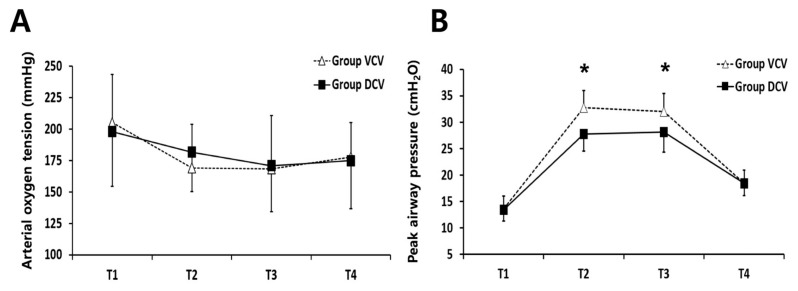
Change in arterial oxygenation (**A**) and peak airway pressure (Ppeak) (**B**) in VCV group and DCV group at four time points. Data are presented as mean values (standard deviation). Abbreviations: VCV = volume-controlled ventilation; DCV = dual-controlled ventilation. Note: T1 = 10 min after anesthesia induction under supine positioning and no pneumoperitoneum; T2 = 30 min after initiation of CO_2_ pneumoperitoneum and Trendelenburg positioning; T3 = 60 min after initiation of CO_2_ pneumoperitoneum and Trendelenburg positioning; T4 = 10 min after CO_2_ desufflation and supine positioning. Note: * From post hoc analysis for multiple comparisons using the Bonferroni correction, Ppeak was significantly lower in DCV group than VCV group at T2 and T3.

**Table 1 jcm-08-02032-t001:** Patient characteristics and perioperative data.

Variables	VCV (*n* = 38)	DCV (*n* = 38)
Age (year)	65.3 (6.2)	64.5 (7.2)
Height (cm)	168.8 (5.0)	166.8 (5.2)
Weight (kg)	68.9 (7.3)	65.8 (7.3)
Body mass index (kg m^−2^)	24.2 (2.8)	23.7 (2.5)
Hypertension (n (%))	16 (42.1)	15 (39.5)
Diabetes mellitus (n (%))	7 (18.4)	3 (7.9)
Coronary artery disease (n (%))	4 (10.5)	2 (5.3)
Duration of anesthesia (min)	157.9 (27.9)	153.4 (22.3)
Duration of surgery (min)	117.7 (24.9)	111.6 (20.9)
Duration of Trendelenburg position (min)	62.4 (18.1)	61.1 (16.3)
Total fluid amounts (mL)	1505.8 (363.6)	1413.7 (321.8)
Colloid amounts (mL)	165.8 (251.8)	155.3 (227.4)
Total urine output (mL)	190.0 (155.0)	167.9 (136.4)
Total blood loss (mL)	407.9 (280.3)	402.6 (245.8)
Total ephedrine amounts (mg)	9.4 (8.7)	7.4 (7.4)
Post anesthesia care unit duration (min)	46.9 (22.5)	45.8 (18.9)
Postoperative hospital stay (day)	2.6 (0.8)	2.5 (0.7)

Data are presented as mean values (standard deviation) and numbers (%). Abbreviations: VCV = volume-controlled ventilation; DCV = dual-controlled ventilation.

**Table 2 jcm-08-02032-t002:** Arterial blood gas analysis and end-tidal carbon dioxide (EtCO_2_) data.

Variables	VCV (*n* = 38)	DCV (*n* = 38)	Estimate (95% CI)	*p*-Value
**pH**				0.803 *
T1	7.43 (0.03)	7.43 (0.03)	−0.005 (−0.019 to 0.009)	>0.999
T2	7.33 (0.04)	7.34 (0.04)	−0.003 (−0.022 to 0.159)	>0.999
T3 ^†^	7.33 (0.04)	7.34 (0.04)	−0.012 (−0.030 to 0.005)	0.509
T4	7.33 (0.04)	7.34 (0.03)	−0.006 (−0.022 to 0.010)	>0.999
**PaO_2_ (mmHg)**				0.149 *
T1	205.2 (38.2)	198.1 (43.5)	7.076 (−11.644 to 25.797)	>0.999
T2	169.2 (34.8)	181.8 (31.6)	−12.687 (−27.874 to 2.500)	0.400
T3 ^†^	168.6 (42.4)	170.9 (36.4)	−2.268 (−20.470 to 15.934)	>0.999
T4	177.8 (27.6)	175.0 (38.3)	2.768 (−12.514 to 18.051)	>0.999
**PaCO_2_ (mmHg)**				0.830 *
T1	32.7 (3.2)	32.4 (3.1)	0.305 (−1.119 to 1.730)	>0.999
T2	44.2 (5.6)	43.8 (6.2)	0.437 (−2.262 to 3.136)	>0.999
T3^†^	44.9 (5.3)	43.5 (6.3)	1.403 (−1.285 to 4.092)	0.905
T4	44.9 (4.6)	43.9 (4.9)	1.039 (−1.118 to 3.197)	>0.999
**EtCO_2_ (mmHg)**				0.716 *
T1	35.0 (2.6)	35.3 (2.2)	−0.368 (−1.459 to 0.723)	>0.999
T2	43.3 (3.7)	42.7 (4.1)	0.658 (−1.126 to 2.442)	>0.999
T3^†^	42.6 (4.1)	42.8 (3.9)	−0.125 (−1.969 to 1.719)	>0.999
T4	43.7 (4.5)	43.5 (3.3)	0.211 (−1.580 to 2.001)	>0.999

Data are presented as mean values (standard deviation). Abbreviations: EtCO_2_ = end-tidal carbon dioxide; VCV = volume-controlled ventilation; DCV = dual-controlled ventilation; CI = confidence interval; PaO_2_ = arterial oxygen tension; PaCO_2_ = arterial carbon dioxide tension. Note: T1 = 10 min after anesthesia induction under supine positioning and no pneumoperitoneum; T2 = 30 min after initiation of CO_2_ pneumoperitoneum and Trendelenburg positioning; T3 = 60 min after initiation of CO_2_ pneumoperitoneum and Trendelenburg positioning; T4 = 10 min after CO_2_ desufflation and supine positioning. Note: * *p*-values of time and group interaction derived from the linear mixed model; **^†^** seventeen patients from VCV group and seventeen patients from DCV group did not reach T3. Among these patients, data was not collected from one patient in DCV group because the robot-assisted procedure time was 31 min.

**Table 3 jcm-08-02032-t003:** Respiratory data.

Variables	VCV (*n* = 38)	DCV (*n* = 38)	Estimate (95% CI)	*p*-Value
**Ppeak (cmH_2_O)**				<0.001 *
T1	13.6 (2.4)	13.4 (2.1)	0.263 (−0.758 to 1.285)	>0.999
T2	32.8 (3.2)	27.8 (3.3)*	5.026 (3.520 to 6.533)	<0.001
T3 ^†^	32.0 (3.4)	28.1 (3.8)*	3.865 (2.209 to 5.520)	<0.001
T4	18.4 (2.5)	18.3 (2.3)	0.131 (−0.967 to 1.230)	>0.999
**Pplat (cmH_2_O)**				<0.001 *
T1	12.4 (2.3)	12.4 (2.0)	0.053 (−0.948 to 1.053)	>0.999
T2	30.2 (4.2)	27.5 (3.4)*	2.711 (0.981 to 4.440)	0.012
T3 ^†^	29.6 (4.0)	27.9 (4.0)	1.713 (−0.119 to 3.546)	0.264
T4	15.8 (3.3)	16.2 (2.5)	−0.368 (−1.695 to 0.958)	>0.999
**Pmean (cmH_2_O)**				0.001 *
T1	4.1 (0.8)	3.8 (0.8)	0.211 (−0.161 to 0.582)	>0.999
T2	8.2 (0.9)	8.6 (1.0)	−0.342 (−0.787 to 0.103)	0.520
T3 ^†^	8.1 (1.0)	8.8 (1.2)	−0.625 (−1.116 to −0.134)	0.053
T4	5.2 (0.8)	5.2 (0.6)	0.026 (−0.293 to 0.345)	>0.999
**TV (mL)**				0.683 *
T1	518.9 (41.3)	493.6 (43.4)	25.3 (6.001 to 44.736)	0.044
T2	528.1 (41.3)	500.7 (47.5)	27.4 (6.981 to 47.650)	0.036
T3 ^†^	530.6 (47.8)	511.9 (51.6)	18.7 (−4.256 to 41.469)	0.436
T4	528.5 (40.6)	506.2 (44.9)	22.2 (2.753 to 41.879)	0.104
**RR (breaths min^−1^)**				>0.999 *
T1	13.4 (1.4)	13.0 (1.4)	0.342 (−0.295 to 0.979)	>0.999
T2	18.6 (3.0)	18.3 (3.5)	0.316 (−1.184 to 1.816)	>0.999
T3 ^†^	18.2 (3.6)	17.7 (3.7)	0.428 (−1.269 to 2.125)	>0.999
T4	19.5 (3.3)	19.2 (2.6)	0.316 (−1.046 to 1.677)	>0.999

Data are presented as mean values (standard deviation). Abbreviations: VCV = volume-controlled ventilation; DCV = dual-controlled ventilation; CI = confidence interval; Ppeak = peak airway pressure; Pplat = plateau airway pressure; Pmean = mean airway pressure; TV = tidal volume; RR = respiratory rate. Note: T1 = 10 min after anesthesia induction under supine positioning and no pneumoperitoneum; T2 = 30 min after initiation of CO_2_ pneumoperitoneum and Trendelenburg positioning; T3 = 60 min after initiation of CO_2_ pneumoperitoneum and Trendelenburg positioning; T4 = 10 min after CO_2_ desufflation and supine positioning. Note: * *p*-values of time and group interaction derived from the linear mixed model; ^†^ seventeen patients from VCV group and seventeen patients from DCV group did not reach T3. Among these patients, data was not collected from one patient in DCV group because the robot-assisted procedure time was 31 min.

**Table 4 jcm-08-02032-t004:** Hemodynamic data.

Variables	VCV (*n* = 38)	DCV (*n* = 38)	Estimate (95% CI)	*p*-Value
**MAP (mmHg)**				0.317 *
T1	74.9 (9.9)	73.2 (11.3)	1.711 (−3.135 to 6.556)	>0.999
T2	81.3 (9.9)	82.2 (10.1)	−0.921 (−5.498 to 3.656)	>0.999
T3 ^†^	74.1 (8.3)	77.4 (9.4)	−3.247 (−7.317 to 0.824)	0.348
T4	69.0 (11.2)	70.4 (9.6)	−1.368 (−6.139 to 3.402)	>0.999
**HR (beats/min)**				0.941 *
T1	73.5 (11.9)	72.4 (10.8)	1.132 (−4.062 to 6.325)	>0.999
T2	72.2 (11.1)	70.6 (11.0)	1.632 (−3.415 to 6.678)	>0.999
T3 ^†^	70.9 (11.4)	70.4 (10.1)	0.436 (−4.525 to 5.397)	>0.999
T4	71.2 (11.1)	70.3 (9.3)	0.816 (−3.874 to 5.505)	>0.999

Data are presented as mean values (standard deviation). Abbreviations: VCV = volume-controlled ventilation; DCV = dual-controlled ventilation; CI = confidence interval; MAP = mean arterial pressure; HR = heart rate. Note: T1 = 10 min after anesthesia induction under supine positioning and no pneumoperitoneum; T2 = 30 min after initiation of CO_2_ pneumoperitoneum and Trendelenburg positioning; T3 = 60 min after initiation of CO_2_ pneumoperitoneum and Trendelenburg positioning; T4 = 10 min after CO_2_ desufflation and supine positioning. Note: * *p*-values of time and group interaction derived from the linear mixed model; ^†^ seventeen patients from VCV group and seventeen patients from DCV group did not reach T3. Among these patients, data was not collected from one patient in DCV group because the robot-assisted procedure time was 31 min.

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
