# Peer review of "Volume-Controlled Versus Dual-Controlled Ventilation during Robot-Assisted Laparoscopic Prostatectomy with Steep Trendelenburg Position: A Randomized-Controlled Trial"

_jcm, 2019, doi:10.3390/jcm8122032_

Round 1
Reviewer 1 Report
MAJOR COMMENTS
Page 2, line 54: …“could reduce negative effects”… The hypothesis should be more concrete in terms of primary and potential secondary outcomes. Oxygenation is a function of end-expiratory lung volume (EELV) and mean airway pressure. Thus, I don’t understand the physiological rationale for changing inspiratory flow pattern and assuming a change in oxygenation. Why didn’t the authors use PEEP in light of in intrabdominal pressure of 15mmHg (about 20cmH20), which is known to be at least partially transferred to pleural pressure and thus will cause compression atelectasis and/or tidal recruitment and collapse? Use of adequate PEEP would have presumably been a higher contribution to lung protection than the inspiratory flow profile. Since PEEP usually improves lung compliance due to alveolar recruitment, the increase in plateau pressure is less than expected. Please discuss. The parameter peak airway pressure is measured in the ventilator and is influenced by inspiratory flow and airway resistance. Thus, changes in inspiratory flow will influence peak airway pressure measured before the ET tube, but not necessarily intraalveolar pressure within the lung. To get reliable values for alveolar pressure, plateau pressure would have been required to measure, which would have required a sufficient inspiratory pause, an inspiratory hold maneuverer, or continuous registration of flow and pressure curves, which would have allowed calculation of tube resistance and tracheal pressure. In addition, the effect of different flow pattern on mean airway pressures would have been interesting. Conclusion: Peak airway pressure is not a relevant parameter in this context. Continuous pressure and flow registration would have added valuable information.MINOR COMMENTS
Nomenclature of ventilatory modes: The authors used VCV with either constant or decelerating flow. The chosen terms auto-flow (trademark of Dräger Medical) and VCV are somewhat problematic. Please consider changes. Abstract: “fever” was assessed postoperatively, not at any time point. Please rephrase. Page 2, line 48: What does “representative” mean in this context.
Author Response
Dear editor,
Thank you for inviting us to revise our manuscript (jcm-633180). We hope that our manuscript has improved through this revision and is now suitable to be considered for possible publication in the Journal of Clinical Medicine.
Our point-by-point responses to the comments raised by you are listed below. We also revised the manuscript and highlighted the revisions in red and rechecked that our manuscript conforms to the Journal’s submission guidelines.
Thank you.
Sincerely,
Min-Soo Kim, MD, PhD.
Reviewer’s comments:
MAJOR COMMENTS
Page 2, line 54: …“could reduce negative effects”… The hypothesis should be more concrete in terms of primary and potential secondary outcomes. Oxygenation is a function of end-expiratory lung volume (EELV) and mean airway pressure. Thus, I don’t understand the physiological rationale for changing inspiratory flow pattern and assuming a change in oxygenation.
Answer) Thank you very much for your considerate comments.
From previous literatures, pressure-controlled ventilation (PCV), with its decelerating flow pattern was considered superior over volume-controlled ventilation (VCV) for decreasing peak airway pressure and the incidence of barotrauma, maintaining uniform distribution of alveolar gas, and improving oxygenation. PCV may also be beneficial to alveolar recruitment by providing high initial flow rates that afforded more rapid and uniform alveolar inflation.
(References: 1) Nichols, D. and S. Haranath, Pressure control ventilation. Crit Care Clin, 2007. 23(2): p. 183-99, viii-ix.; 2) Al-Saady, N. and E.D. Bennett, Decelerating inspiratory flow waveform improves lung mechanics and gas exchange in patients on intermittent positive-pressure ventilation. Intensive Care Med, 1985. 11(2): p. 68-75.; 3) TuÄŸrul M, Camci E, Karadeniz H, Sentürk M, Pembeci K, Akpir K. Comparison of volume controlled with pressure controlled ventilation during one-lung anaesthesia. Br J Anaesth 1997;79:306-10.;4) Prella M, Feihl F, Domenighetti G. Effects of short-term pressure controlled ventilation on gas exchange, airway pressures, and gas distribution in patients with acute lung injury/ARDS. Comparison with volume-controlled ventilation. Chest 2002;122:1382-8.)
For these reasons, PCV has been proposed as an alternative to VCV in patients with reduced lung compliance, such as ICU patients with acute respiratory syndrome or obese surgical patients, to achieve adequate oxygenation and normocapnia.
(References: 1) Mercat A, Graini L, Teboul JL, Lenique F, Richard C. Cardiorespiratory effects of pressure-controlled ventilation with and without inverse ratio in the adult respiratory distress syndrome. Chest 1993; 104: 871–5; 2) Prella M, Feihl F, Domenighetti G. Effects of short-term pressure-controlled ventilation on gas exchange, airway pressures, and gas distribution in patients with acute lung injury/ARDS: comparison with volume-controlled ventilation. Chest 2002; 122: 1382–8; 3) Ogunnaike BO, Jones SB, Jones DB, Provost D, Whitten CW. Anesthetic considerations for bariatric surgery. Anesth Analg 2002; 95: 1793–805).
However, PCV has the limitation of variable tidal volume and minute ventilation secondary to the patient’s lung compliance and airway resistance changes. In robotic surgery, surgical manipulations, CO2 pneumoperitoneum and steep Trendelenburg position usually alter patient’s lung compliance greatly, so PCV may be difficult to maintain an adequate and constant tidal volume (and hence, minute volume). Autoflow, one type of dual-controlled ventilation (DCV) and also known as pressure-controlled ventilation with volume guaranteed (PCV with VG), has the advantage of ensuring tidal volume while having a decelerating flow pattern. Thus, we hypothesized that application of Autoflow which ensures tidal volume with decelerating flow pattern could be beneficial to improve oxygenation and prevent barotrauma under adverse respiratory changes during RALRP and the decline in pulmonary function in the elderly.
We revised our sentence about hypothesis to clarify its meaning as follows.
Original sentence: We hypothesized that the use of AF could reduce negative effects of CO2 pneumoperitoneum and steep Trendelenburg position on respiratory system during RALRP.
Revised sentence: We hypothesized that application of DCV which ensures tidal volume with decelerating flow pattern could be beneficial to improve oxygenation and prevent barotrauma by reducing negative effects of CO2 pneumoperitoneum and steep Trendelenburg position on respiratory system during RALRP.
We absolutely agree that oxygenation is a function of end-expiratory lung volume (EELV) and mean airway pressure. Oxygenation may be affected by various factors such as distribution of alveolar gas, initial flow rates and mean airway pressure. In this study, Autoflow (dual-controlled ventilation) did not improve oxygenation compared with VCV. In addition, there were no significant differences in mean airway pressure between VCV and Autoflow (dual-controlled ventilation). Considering that mean airway pressure is closely related to mean alveolar pressure, we have inferred that no difference in mean airway pressure might result in lack of improvement in oxygenation between VCV and Autoflow (dual-controlled ventilation). We have described regarding mean airway pressure and oxygenation in the discussion section (paragraph 3 and 4).
Why didn’t the authors use PEEP in light of in intrabdominal pressure of 15mmHg (about 20cmH20), which is known to be at least partially transferred to pleural pressure and thus will cause compression atelectasis and/or tidal recruitment and collapse? Use of adequate PEEP would have presumably been a higher contribution to lung protection than the inspiratory flow profile. Since PEEP usually improves lung compliance due to alveolar recruitment, the increase in plateau pressure is less than expected. Please discuss.
Answer) We also agree with your comments regarding PEEP. In this study, we did not use PEEP to purely compare the two ventilation modes, because PEEP itself could affect oxygenation and airway pressure.
Also, there are several previous studies that compared ventilatory modes without use of PEEP.
1) Choi EM, Na S, Choi SH, An J, Rha KH, Oh YJ. Comparison of volume-controlled and pressure-controlled ventilation in steep Trendelenburg position for robot-assisted laparoscopic radical prostatectomy. Journal of clinical anesthesia. 2011;23(3):183-8.
2) Song SY, Jung JY, Cho MS, Kim JH, Ryu TH, Kim BI. Volume-controlled versus pressure-controlled ventilation-volume guaranteed mode during one-lung ventilation. Korean journal of anesthesiology. 2014;67(4):258-63.
3) Balick-Weber CC, Nicolas P, Hedreville-Montout M, Blanchet P, Stephan F. Respiratory and haemodynamic effects of volume-controlled vs pressure-controlled ventilation during laparoscopy: a cross-over study with echocardiographic assessment. British journal of anaesthesia. 2007;99(3):429-35.
4) Tugrul M, Camci E, Karadeniz H, Senturk M, Pembeci K, Akpir K. Comparison of volume controlled with pressure controlled ventilation during one-lung anaesthesia. British journal of anaesthesia. 1997;79(3):306-10.
5) Assad OM, El Sayed AA, Khalil MA. Comparison of volume-controlled ventilation and pressure-controlled ventilation volume guaranteed during laparoscopic surgery in Trendelenburg position. Journal of clinical anesthesia. 2016;34:55-61.
Considering the possibility of criticism that the use of PEEP might affect the assessment and comparison of the two ventilatory mode, we determined not to apply PEEP on the basis of these previous studies.
We added the sentences regarding the use of PEEP in the discussion section as a limitation of this study.
Added sentences: Lastly, patients were ventilated without applying PEEP to exclusively compare VCV and DCV. The primary effect of PEEP consists of an increase in functional residual capacity by the expansion and stabilization of collapsed alveoli. Therefore, the application of adequate PEEP may result in different contribution to oxygenation and lung compliance during CO2 pneumoperitoneum and steep Trendelenburg positioning. Thus, further studies are needed to identify the role of the PEEP and its contribution to oxygenation under each ventilatory mode.
The parameter peak airway pressure is measured in the ventilator and is influenced by inspiratory flow and airway resistance. Thus, changes in inspiratory flow will influence peak airway pressure measured before the ET tube, but not necessarily intraalveolar pressure within the lung. To get reliable values for alveolar pressure, plateau pressure would have been required to measure, which would have required a sufficient inspiratory pause, an inspiratory hold maneuverer, or continuous registration of flow and pressure curves, which would have allowed calculation of tube resistance and tracheal pressure. In addition, the effect of different flow pattern on mean airway pressures would have been interesting. Conclusion: Peak airway pressure is not a relevant parameter in this context. Continuous pressure and flow registration would have added valuable information.
Answer) Thank you for your comments. In this study, primary outcome is arterial oxygenation. We also collected plateau and mean airway pressure as well as peak airway pressure as respiratory data. As you mentioned, each airway pressure has its own meaning in terms of respiratory mechanics, so we compared and analyzed these pressures with equal importance. We used inspiratory pause, i.e. the plateau time set to 10% of total inspiration time for plateau pressure. We have described the analysis results of three kinds of airway pressure in the result section. We have also described the meaning and consideration of mean airway pressure in the discussion section.
We revised the conclusion as follows, according to your comments.
Original sentences: In conclusion, our study did not support the application of AF for improving oxygenation. However, this study showed that AF could lower Ppeak without producing hemodynamic instability during CO2 pneumoperitoneum with steep Trendelenburg positioning.
Revised sentences: In conclusion, our study did not support the application of DCV for improving oxygenation. However, this study showed that DCV have a potential to lower Ppeak or Pplat without producing hemodynamic instability during CO2 pneumoperitoneum with steep Trendelenburg positioning.
MINOR COMMENTS
Nomenclature of ventilatory modes: The authors used VCV with either constant or decelerating flow. The chosen terms auto-flow (trademark of Dräger Medical) and VCV are somewhat problematic. Please consider changes.
Answer) We modified the term “Autoflow (AF)” to “dual-controlled ventilation (DCV)” according to your recommendation.
Abstract: “fever” was assessed postoperatively, not at any time point. Please rephrase.
Answer) We modified the sentence according to your recommendation.
Original sentence: Postoperative fever was more common in group VCV (12%) than in group AF (3%) (P = 0.022).
Modified sentence: Postoperative fever occurring within the first 2 days after surgery was more common in group VCV (12%) than in group DCV (3%) (P = 0.022).
Page 2, line 48: What does “representative” mean in this context.
Answer) In this context, “representative” is used to mean “commonly performed”. Thus, we modified the sentence to clarify its meaning.
Original sentence: Robot-assisted laparoscopic radical prostatectomy (RALRP) is a representative robotic surgery using CO2 pneumoperitoneum and steep Trendelenburg positioning,
Modified sentence: Robot-assisted laparoscopic radical prostatectomy (RALRP) is a commonly performed robotic surgery using CO2 pneumoperitoneum and steep Trendelenburg positioning,

Reviewer 2 Report
I have read the manuscript from Park and colleagues with interest. They compared volume controlled ventilation mode to VC+autoflow option. Although the autoflow option did not improve oxygenation, The study was able to show a decrease of Ppeak and postoperative fever rate in the group with autoflow option. In general the manuscript was well written and the study design and analysis methods were sound. I have only a few suggestions:
Page 3, line 108-114, it would be better to have a flowchart explaining the time points and sequences.
Page 4, line 152, could the author specify what protocol violations there were?
Figure 1, I don't think the items with n=0 were adding any information. Please delete those n=0 items in figure 1
In tables 2-4 the first variables were in bold, which were not necessary.
Page 8, line 252, I think the word decelerating might not be suitable to describe the flow pattern. It is more like an exponential decay.
Page 9, from line 280 to 285: ” Atelectasis has been known as the primary cause of early postoperative fever [28,30].” And “there was no obvious evidence supporting atelectasis as the main cause of early postoperative fever [28]” these two statements were contradictory. Moreover, they were quoted from the same reference [28].
Author Response
Dear editor,
Thank you for inviting us to revise our manuscript (jcm-633180). We hope that our manuscript has improved through this revision and is now suitable to be considered for possible publication in the Journal of Clinical Medicine.
Our point-by-point responses to the comments raised by you are listed below. We also revised the manuscript and highlighted the revisions in red and rechecked that our manuscript conforms to the Journal’s submission guidelines.
Thank you.
Sincerely,
Min-Soo Kim, MD, PhD.
Reviewer’s comments:
I have read the manuscript from Park and colleagues with interest. They compared volume controlled ventilation mode to VC+autoflow option. Although the autoflow option did not improve oxygenation, The study was able to show a decrease of Ppeak and postoperative fever rate in the group with autoflow option. In general the manuscript was well written and the study design and analysis methods were sound. I have only a few suggestions:
Page 3, line 108-114, it would be better to have a flowchart explaining the time points and sequences.
Answer) We added a flowchart (figure 1), according to your recommendation. Also, we notified that the term “AF” was changed to “DCV (dual-controlled ventilation)” according to the other reviewer’s recommendation.
Page 4, line 152, could the author specify what protocol violations there were?
Answer) Two patients from each group were withdrawn because of lowering intra-abdominal pressure (15 mmHg -> 10 mmHg) due to high peak airway pressure and errors in setting of FiO2, tidal volume and ventilatory mode.
We revised the sentence as follows:
Original sentence: Two patients from each group were withdrawn from the analysis because of high Ppeak and protocol violations
Revised sentence: Two patients from each group were withdrawn from the analysis because of high Ppeak and protocol violations; intra-abdominal pressure had to be reduced from 15 mmHg to 10 mmHg due to excessively elevated Ppeak and errors occurred in the setting of FiO2 and tidal volume on the ventilator machine.
Figure 1, I don't think the items with n=0 were adding any information. Please delete those n=0 items in figure 1
Answer) We revised the figure 2 (patient flow chart) according to your recommendation.
In tables 2-4 the first variables were in bold, which were not necessary.
Answer) We revised the tables according to your recommendation.
Page 8, line 252, I think the word decelerating might not be suitable to describe the flow pattern. It is more like an exponential decay.
Answer) We replaced the phrase “decelerating” to “exponential decay”, according to your recommendation.
Page 9, from line 280 to 285: ” Atelectasis has been known as the primary cause of early postoperative fever [28,30].” And “there was no obvious evidence supporting atelectasis as the main cause of early postoperative fever [28]” these two statements were contradictory. Moreover, they were quoted from the same reference [28].
Answer) The Introduction part of reference [28] stated that most surgical textbooks have adopted the concept that atelectasis is the most common cause of early postoperative fever (EPF), some claiming that atelectasis “is responsible for over 90% of febrile episodes during that period” (the first 48 h after operation). In addition, the same reference concluded that the available evidence regarding the association of atelectasis and fever is scarce. Thus, the statements were quoted from the same reference [28] although they were contradictory. We removed reference [28] from first sentence considering the conclusion of the reference [28].

Reviewer 3 Report
Specific comments
|
Comment |
Reference |
Comment |
|
#1 |
Line 56 |
“Respiratory mechanics” are not enough reflected in the methods section, see comment #3. |
|
#2 |
Section 2.3 Ventilation management |
Please add more detailed ventilator settings during the CO2-peritoneum and steep Trendelenburg position (PEEP? FIO2?). If these were identical with the settings prior to skin incision, add the note that settings were unchanged despite of changing the mode according to the randomization result. Have there been rescue strategies? |
|
#3 |
Section 2.4 Data collection |
In the hypothesis, you name „lung mechanics“ as an endpoint. However, both the methods section and the results section report airway pressures, which may be related to lung mechanics, but do not represent lung mechanics in the narrower sense. While aiming to investigate lung mechanics, compliance and driving pressure, as well as resistance should be reported. Differences in both peak and plateau but not mean airway pressure may suggest differences in the resistance. While resistance is closely related to airway flow, corresponding results would have been interesting. Please either consider reporting additional data (compliance, resistance, driving pressure) or changing the wording (“airway pressures” instead of “lung mechanics”). The total duration of the follow-up time frame is not defined in the methods section. Therewith, it is unclear how long you followed the patients (e.g. one, two, three, or more postoperative days) and which timeframe has been defined regarding postoperative fever. |
|
#4 |
Figure 1 |
Please use American English consequently (“randomized” in stead of “randomised” in Figure 1). In Figure 1, some “n=” are written in italics and one is not. Please adjust. |
|
#5 |
Figure 2 |
What is shown in Figure 2A/B? Is it mean and standard deviation? Please add information to the legend below the figure. |
|
#6 |
Table 3 |
You may add tidal volume to the table 3. |
|
#7 |
Line 204 |
Respiratory complications have not been defined in the manuscript. Please define respiratory complications in the methods section. |
|
#8 |
Line 258 |
Please check the sentence for errors (“…and functional limitation and may be more…”). |
|
#9 |
Lines 286-288 |
Please explain how you assume that the ventilator modes may have affected the surgical stress and the inflammatory response. |
|
#10 |
Lines 293-296 |
Why is this a specific limitation of your specific study? |
|
#11 |
Lines 299-300 |
This fact is already included in your first limitation and represents a repetition. |
|
#12 |
Lines 300-301 |
This is not a limitation of your study. |

Author Response
Dear editor,
Thank you for inviting us to revise our manuscript (jcm-633180). We hope that our manuscript has improved through this revision and is now suitable to be considered for possible publication in the Journal of Clinical Medicine.
Our point-by-point responses to the comments raised by you are listed below. We also revised the manuscript and highlighted the revisions in red and rechecked that our manuscript conforms to the Journal’s submission guidelines.
Thank you.
Sincerely,
Min-Soo Kim, MD, PhD.
Reviewer’s comments:
|
#1 |
Line 56 |
“Respiratory mechanics” are not enough reflected in the methods section, see comment #3. |
Answer) Thank you for your comment. According to your recommendation in comment #3, we changed the wording “airway pressures” instead of “lung mechanics. Also, we notified that the term “AF” was changed to “DCV (dual-controlled ventilation)” according to the other reviewer’s recommendation.
Original sentence: Therefore, we designed this study to compare the effects of AF and VCV mode on oxygenation and respiratory mechanics in patients undergoing RALRP.
Revised sentences: Therefore, we designed this study to compare the effects of DCV and VCV mode on oxygenation and airway pressure in patients undergoing RALRP.
|
#2 |
Section 2.3 Ventilation management |
Please add more detailed ventilator settings during the CO2-peritoneum and steep Trendelenburg position (PEEP? FIO2?). If these were identical with the settings prior to skin incision, add the note that settings were unchanged despite of changing the mode according to the randomization result. Have there been rescue strategies? |
Answer) We did not change ventilatory settings despite of changing the mode according to the group allocation. We added a flowchart (figure 1) to clarify the ventilation management according to other reviewer’s recommendation. In addition, we revised the sentence as follows.
Original sentence: Immediately after Trendelenburg positioning with CO2 pneumoperitoneum, ventilator mode in the group AF was changed from VCV to AF.
Revised sentence: Immediately after Trendelenburg positioning with CO2 pneumoperitoneum, ventilator mode in the group DCV was changed from VCV to DCV without changes in settings of other ventilatory parameters.
According to our study protocol, patients were withdrawn from the study if oxygen desaturation (SpO2 < 95%) or increase of Ppeak > 40 cmH2O occurred. In these situations, we discussed with surgeons and applied high concentrations of oxygen, reduced intra-abdominal pressure, or performed lung recruitment maneuver.
|
#3 |
Section 2.4 Data collection |
In the hypothesis, you name „lung mechanics“ as an endpoint. However, both the methods section and the results section report airway pressures, which may be related to lung mechanics, but do not represent lung mechanics in the narrower sense. While aiming to investigate lung mechanics, compliance and driving pressure, as well as resistance should be reported. Differences in both peak and plateau but not mean airway pressure may suggest differences in the resistance. While resistance is closely related to airway flow, corresponding results would have been interesting. Please either consider reporting additional data (compliance, resistance, driving pressure) or changing the wording (“airway pressures” instead of “lung mechanics”). The total duration of the follow-up time frame is not defined in the methods section. Therewith, it is unclear how long you followed the patients (e.g. one, two, three, or more postoperative days) and which timeframe has been defined regarding postoperative fever. |
Answer) According to your recommendation, we changed the wording “airway pressures” instead of “lung mechanics”. We agreed that differences in airway pressures (Ppeak, Pplat, not Pmean) is related to airway flow and resistance. In this study, peak airway pressure (Ppeak), plateau airway pressure (Pplat), mean airway pressure (Pmean), EtCO2, RR, and compliance were collected by reading the values displayed on the monitor of the Primus anesthetic workstation (Dräger, Lübeck, Germany) once in each time point. However, inspiratory resistance could not be directly collected from anesthesia machine, but only be inferred from other variables. In addition, there were no significant differences in compliance between the groups, although we did not describe about the result of compliance. Therefore, we changed the wording “lung mechanics” to “airway pressure”.
We revised the sentences in the method section about the follow-up time frame and respiratory complications
Original sentence: We also collected data regarding the duration of surgery, pneumoperitoneum, anesthesia, post-anesthesia care unit (PACU) stay, and hospital stay; intraoperative blood loss, fluid intake, urine output, and amount of administered vasopressors; and presence of postoperative fever (> 38.0°C).
Revised sentence: We also collected data regarding the duration of surgery, pneumoperitoneum, anesthesia, post-anesthesia care unit (PACU) stay, and hospital stay; intraoperative blood loss, fluid intake, urine output, amount of administered vasopressors, and respiratory complications such as pneumothorax and desaturation events during intraoperative and recovery period; and respiratory complications and presence of postoperative fever (> 38.0°C) within two postoperative days.
|
#4 |
Figure 1 |
Please use American English consequently (“randomized” in stead of “randomised” in Figure 1). In Figure 1, some “n=” are written in italics and one is not. Please adjust. |
Answer) We revised the figure 2 (patient flow chart). According to other reviewer’s recommendation (I don't think the items with n=0 were adding any information. Please delete those n=0 items in figure 1), we deleted n=0 items in figure 2.
|
#5 |
Figure 2 |
What is shown in Figure 2A/B? Is it mean and standard deviation? Please add information to the legend below the figure |
Answer) We added the following sentence.
Added sentence: Data are presented as mean (standard deviation).
|
#6 |
Table 3 |
You may add tidal volume to the table 3 |
Answer) We added results of tidal volume in Table 3.
|
#7 |
Line 204 |
Respiratory complications have not been defined in the manuscript. Please define respiratory complications in the methods section. |
Answer) As mentioned above in comment #3, we revised the sentences as follows.
Original sentence: We also collected data regarding the duration of surgery, pneumoperitoneum, anesthesia, post-anesthesia care unit (PACU) stay, and hospital stay; intraoperative blood loss, fluid intake, urine output, and amount of administered vasopressors; and presence of postoperative fever (> 38.0°C).
Revised sentence: We also collected data regarding the duration of surgery, pneumoperitoneum, anesthesia, post-anesthesia care unit (PACU) stay, and hospital stay; intraoperative blood loss, fluid intake, urine output, amount of administered vasopressors, and respiratory complications such as pneumothorax and desaturation events during intraoperative and recovery period; and respiratory complications and presence of postoperative fever (> 38.0°C) within two postoperative days.
|
#8 |
Line 258 |
Please check the sentence for errors (“…and functional limitation and may be more…”). |
Answer) We corrected the sentence. We apologize for the confusion.
Original sentence: Surgical conditions or patient factors such as age, underlying disease, nutrition status, and functional limitation and may be more important
Revised sentence: Surgical conditions or patient factors such as age, underlying disease, nutrition status, and functional limitation may be more important
|
#9 |
Lines 286-288 |
Please explain how you assume that the ventilator modes may have affected the surgical stress and the inflammatory response. |
Answer) In this current study, DCV was associated with a significantly lower incidence of postoperative fever. Surgical stress and the accompanying increase in circulating inflammatory cytokines may be associated with early postoperative fever. Carbon dioxide (CO2) insufflation is used during a laparoscopy procedure, but this may affect stress responses. The study about impact of a lung-protective ventilatory strategy on systemic and pulmonary inflammatory responses during laparoscopic surgery was performed recently. (reference: Kokulu, S.; Günay, E.; Baki, E.D.; Ulasli, S.S.; Yilmazer, M.; Koca, B.; Arıöz, D.T.; Ela, Y.; Sivaci, R.G. Impact of a lung-protective ventilatory strategy on systemic and pulmonary inflammatory responses during laparoscopic surgery: Is it really helpful? Inflammation 2015, 38, 361-367.) From the lower incidence of early postoperative fever in DCV, we may infer the difference in surgical stress or inflammation according to the ventilation mode, but further study should be needed.
We have added the study of Kokulu et al. as our reference #31.
|
#10 |
Lines 293-296 |
Why is this a specific limitation of your specific study? |
Answer) Thank you for your thoughtful comment. We have described these sentences as a common limitation derived from various types of dual-controlled ventilation and difference of type of surgery, surgical position and patient characteristics in this kind of researches. However, we removed these sentences according to your comments.
|
#11 |
Lines 299-300 |
This fact is already included in your first limitation and represents a repetition. |
Answer) We removed this sentence regarding outcome investigator.
|
#12 |
Lines 300-301 |
This is not a limitation of your study. |
Answer) We removed this sentence.

Round 2
Reviewer 1 Report
Thank you for addressing the comments.
Reviewer 3 Report
Dear authors,
Thank you for submitting a revised version of your manuscript. I still have some minor comments:
Line 108: Please check English and change to "Study flow chart is shown..."
Line 128: Please write "I:E, inspiratory to expiratory time..."
Figure 2: In the figure, all "n" are written in italics, but not the "n" in the lower right corner. Please write all "n" in italics.
Thank you.